# How Do Player Substitutions Influence Men's UEFA Champions League Soccer Matches?

Blanca Iglesias [1], Juan M. García-Ceberino [2,3,*] , Javier García-Rubio [1,4] and Sergio J. Ibáñez [1,4,*]

1. Optimization of Training and Sports Performance Research Group (GOERD), Faculty of Sports Science, University of Extremadura, 10003 Cáceres, Spain
2. Universidad de Extremadura, Facultad de Educación y Psicología, Avenida de Elvas s/n, 06006 Badajoz, Spain
3. Faculty of Education, Psychology and Sports Science, University of Huelva, 21007 Huelva, Spain
4. Faculty of Sports Science, University of Extremadura, 10003 Cáceres, Spain
* Correspondence: jmanuel.jmgc@gmail.com (J.M.G.-C.); sibanez@unex.es (S.J.I.)

**Abstract:** Coaches' player substitution strategies can change the tactical behavior and the final result of matches. This empirical study aims to describe the relationship/association of player substitution variables with the results of men's UEFA Champions League matches during the 2018–2019 season. A total of 125 matches were analyzed using an ad hoc observation sheet created for this purpose. To measure the degree and strength of association between the variables studied, Chi-square and Cramer's V tests were used, respectively. In turn, the Adjusted Standardized Residuals from the contingency tables were calculated to detect patterns of association. Likewise, a decision tree, in particular, the CHAID method, was used to predict and identify interactions. Player substitutions affect the final result and the findings prove it. An own team's goal after 5–10 minutes of player substitution was win-related (positive impact) (90.40%, *ASRs* = 10.40), while an opposing team's goal after 5–10 minutes of player substitution was loss-related (negative impact) (90.30%, *ASRs* = 10.30). Regardless of the match status, the positive impact increased the winning percentage. Furthermore, the match status was postulated as an indicator of the need to make player substitutions. It showed that player substitutions could determine the final result when teams were tying. On the other hand, the match location was not a differentiating factor between winning and losing teams, although the winning percentage was somewhat higher for home teams. Coaches could use this information to establish player substitution strategies that would allow them to perform at their best.

**Keywords:** final result; goal; home advantage; match status; notational analysis; time

## 1. Introduction

The main function of the soccer coach during a match is to manage the various changing situations that occur in order to win, maintain an advantage or recover from a disadvantage [1]. The analysis of player substitutions made by coaches is an emerging research topic in high-performance soccer [2–4].

In a substitution, one player who is on the bench replaces a player who started the match. The substituted player may not re-enter the field of play [5]. Soccer player substitutions can be made for different reasons, such as a player being injured, being fatigued or having a yellow card or to modify tactical systems [6]. In soccer, there is limited number of player substitutions. In this regard, until the advent of the COVID-19 pandemic, the regulations allowed a maximum of three player substitutions during an official match and one additional player substitution in the event of extra time, in accordance with rule 3 of the 2014 International Football Association Board (IFAB) regulations. However, at the beginning of this pandemic, the IFAB established a temporary rule (extended until December 2022) allowing soccer teams to make up to five player substitutions [7]. Despite

this, Mota et al. [8], in relation to the COVID-19 pandemic, have reflected on the importance of further increasing the number of player substitutions to reduce the likelihood of injury.

Elite soccer is a team sport where positive results are paramount and the competition demands that players play at the highest level [9]. It is also an intermittent sport that requires the capacity of players to repeatedly perform high-intensity actions [10]. Therefore, most of the research conducted on player substitutions has focused on external [11] and internal [12] intensities, especially in the distances covered at different speed ranges [2,3,13]. During soccer matches, the physical and technical performance of players decreases, and the current evidence [2] suggests that substitute players report a greater percentage of time spent in distances covered at higher intensity compared to players playing the entire match. Thus, player substitutions are justified as an effective strategy to reduce the effects of team fatigue [14]. However, the main reason for making player substitutions during soccer matches is not to mitigate the fatigue of the players on the field, but to change the tactical behavior of the teams [4]. Along this line, Gomez et al. [15] suggest that the tactical behavior and playing style of teams during soccer matches change after the player substitutions. Thus, variability of tactical systems leads to high performance and competitive advantage [16]. Match congestion, including during national league and major cup competitions, is another factor affecting player substitutions in modern soccer [17]. It is impossible to have an accurate record of the teams' national competition calendar in its different competitions (regular league; cup format, knockout competitions), time of the season and/or players' participation in international competitions with their countries. Therefore, in this study, only the calendar of the competition analyzed was taken into account, although it is clear that the aforementioned factors influence the use of substitutions.

The analysis of different situational variables has expanded the scientific knowledge on this research topic. The variables selected for analysis in this study are those most commonly used in this type of research and describe the phase of the competition (competition stage), the place where the match is played (match location), partial result of the match (match status) and the result of the match (final result).

### 1.1. The Competition Stages (CS)

Regarding elite soccer competitions, Cerda et al. [18] analyzed the performance of players in the group and knockout stages of the 2018 FIFA World Cup together. This competition involves different elite players from all continents, and it shares the development of these stages of competition with the UEFA Champions League (UCL). There is previous research that studied these stages of competition in the 2014 [19] and 2018 [20] FIFA World Cups, focusing only on one of them and identifying performance indicators. With respect to the UCL, other previous studies also analyzed the group stage [21], knockout stage [22] and both stages [23].

### 1.2. The Match Location (ML) or Home Advantage

The ML has been studied in national leagues such as the Spanish League, and the evidence reports that home teams make player substitutions earlier than visiting teams [15]. Furthermore, in the Spanish League, Fernández-Cortés et al. [24] indicated home team advantages in terms of shots, fouls, offsides and goalkeeper saves. Regarding the UCL, the ML is an indicator that differs between the winning and losing teams in the group stage [21]. On the contrary, playing the second leg at home or the first leg away does not report statistically significant differences on the final result in the knockout stage [22].

### 1.3. Match Status (MS) and Time at Which the Coaches Make Player Substitutions (PST)

In the national leagues (English Premier League, Italian Serie A or Spanish League), when teams are losing during the matches, the first player substitution is usually made around the 58th minute, the second player substitution around the 73rd minute, and the third player substitution around the 79th minute. In contrast, when teams are tying or winning, the moment at which player substitutions are made does not matter [25].

Regarding the Spanish League, when teams are losing, they make player substitutions earlier than teams that are tying or winning. Moreover, most of the first and second player substitutions are made in the period from the 61st to the 90th minute, while the third player substitution is made in the period from the 76th to the 90th minute [15]. Similar results have been reported in the UCL; whereby, when teams are losing, the first player substitution is usually made before the 53rd minute, the second before the 71st minute, and the third before the 80th minute [5].

### 1.4. The Final Result (FR) of the Match

In the Spanish League, teams that score first, whether at home or away, win the most matches [26]. In this league, Gomez et al. [15] indicated that a first player substitution made between the 56th and 70th minutes by the losing team is associated with a higher probability of ending the match tying, and a second player substitution made between the 70th and 75th minutes by these losing teams is associated with a higher probability of evening the score. In this line of research, a recent study [27] concluded that soccer teams increase the probability of scoring a goal after the first and second player substitution, and they decrease the probability of scoring a goal after all player substitutions made by the opposing team. In turn, Silva and Swartz [28] pointed out that special care should be taken with early player substitutions since, if a player is injured after all permitted player substitutions have been made, teams must continue with one less player.

### 2. Study Purpose

It has been shown that coaches' player substitution strategies, in relation to the timing of substitutions, can change the tactical behavior and, as a consequence, the final result of soccer matches [15]. It should be noted that empirical research to date on the association between player substitutions and the FR of soccer matches is quite limited [27]. Thus, the study's purpose was to describe the relationship/association of player substitution variables with the results of men's UCL matches during the 2018–2019 season. It was divided into three specific objectives: (1) to characterize the relationship between the time at which player substitutions are made and the time at which goals are scored; (2) to analyze the associations between the situational variables named CS, ML, MS, PST, repercussion of the player substitutions on goals and FR; and (3) to predict the influence of the situational variables on the FR.

### 3. Materials and Method

#### 3.1. Study Design

A descriptive study with a predictive associative strategy was conducted using an arbitrary observation code [29] in order to explore the relationships between the variables studied and predict/explain their behavior.

#### 3.2. Sample

All men's UCL matches of the 2018–2019 season were analyzed (*N* = 125). Figure 1 shows a classification of the analyzed matches according to CS.

#### 3.3. Variables and Instrument

The variables used in the analyses, as well as their categories, were as follows:

- Competition Stage (CS): (1) group stage; and (2) knockout stage.
- Match Location (ML): (1) home team; (2) visiting team; and (3) neutral team.
- Match Status (MS): Result at the time the coaches made any player substitution: (1) tying; (2) winning; and (3) losing.
- Player Substitutions Time (PST): Minute in which the player substitutions were made: (0) no player substitution; (1) minute 0:00 to end of 1st half; (11) minute 45:00 to 50:00; (12) minute 50:01 to 55:00; (13) minute 55:01 to 60: 00; (14) minute 60:01 to 65:00; (15) minute 65:01 to 70:00; (16) minute 70:01 to 75: 00; (17) minute 75:01 to 80:00;

(18) minute 80:01 to 85:00; (19) minute 85:01 to 90:00; and (20) minute 90:01 to end of 2nd half. The extra time category was not included in the analysis because it was not statistically significant.

- Goal Time (GT): Minute in which a goal was scored. It presents the same categories as the PST variable.
- Repercussion of the Player Substitutions on Goals (RPSG): A classification was established that takes into account the period of time that elapsed from the player substitution until a goal was scored. This classification uses 5-minute intervals (Table 1). This variable was calculated as: RPSG = GT − PST.
- Impact of RPSG (IRPSG): (1) positive impact (own team goal after a player substitution); (2) negative impact (goal by the opposing team after a player substitution); (99) no impact.
- Final Result (FR) of the match: (1) tie; (2) win; and (3) loss.

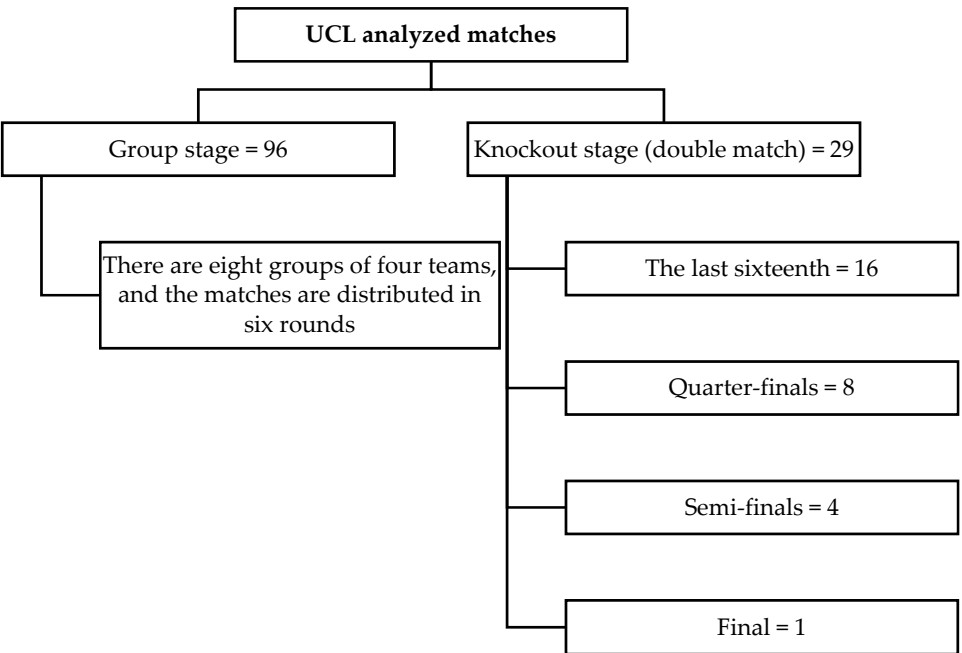

**Figure 1.** Classification of the analyzed UCL matches according to CS.

**Table 1.** Interpretation of the RPSG variable.

| RPSG Interpretation | Time between PST–GT (Minute) |
|---|---|
| Immediate | 0:00–5:00 |
| Short-term | 5:01–10:00 |
| Medium-term | 10:01–15:00 |
| Medium–long-term | 15:01–20:00 |
| Long-term | 20:01 onwards |
| No repercussion | No goal after player substitution |

Note: RPSG = Repercussion of the Player Substitutions on Goals; PST = Player Substitutions Time; GT = Goal Time.

The instrument used was an "ad hoc" observation sheet created in a Microsoft Excel spreadsheet for Windows.

### 3.4. Procedure

An observation sheet was created according to the study interests, in which the variables studied were inserted. In addition, the different categories of each variable were defined. The data were collected making use of this ad hoc observation sheet. Thereafter,

all UCL matches of the 2018–2019 season were analyzed using the free Flashcore ES App (company Livesport Media Ltd., San Gwann, Malta).

Finally, the data collected on the observation sheet were imported into the statistical program SPSS v.25 (IBM Corp. Released 2017. IBM SPSS Statistics for Windows, Version 25, IBM Corp, Armonk, NY, USA) for analysis, in order to interpret the different values.

*3.5. Statistical Analysis*

Firstly, the Chi-square ($X^2$) test was calculated to indicate the degree of relationship between the variables studied. Likewise, Cramer's V coefficient indicated the strength of association between the variables. In this regard, Crewson [30] established different levels of association depending on the obtained value: <0.100 (small), 0.100–0.299 (low), 0.300–0.499 (moderate) and >0.500 (high).

Then, the Adjusted Standardized Residuals (*ASRs* > |1.96|) from the contingency tables were calculated to detect patterns of association, indicating more or less cases than expected [31]. The *ASRs* also provided the frequencies and percentages resulting from crossing the categories of the different variables selected.

Lastly, a decision tree was used to predict and identify interactions. In particular, the CHAID method was used, which made it possible to automatically detect interactions using the Chi-square test. This method indicated which predictor variable had the greatest interaction with the dependent variable (FR). Likewise, it established a partition of nodes in which each node showed the percentage for each category of the dependent variable. Thus, category merging occurred when the categories of each predictor were not significantly different with respect to the dependent variable [32]. The decision tree has also been used in professional basketball [33].

## 4. Results

*4.1. RPSG, FR and IRPSG Variables*

Table 2 shows the degree of relationship between the RPSG, FR and IRPSG variables. There are significant associations between the three variables, regardless of RPSG, with a moderate and high strength association.

**Table 2.** Degree of relationship between the RPSG, FR and IRPSG variables.

| Relationship | $X^2$ | df | p-Value | Vc | Strength |
|---|---|---|---|---|---|
| Immediate * FR * IRPSG | 17.515 | 2 | 0.00 * | 0.44 | Moderate |
| Short-term * FR * IRPSG | 121.929 | 2 | 0.00 * | 0.75 | High |
| Medium-term * FR * IRPSG | 80.982 | 2 | 0.00 * | 0.70 | High |
| Medium–long-term * FR * IRPSG | 41.657 | 2 | 0.00 * | 0.70 | High |
| Long-term * FR * IRPSG | 121.689 | 2 | 0.00 * | 0.75 | High |

Note: $X^2$ = Chi-square test; *df* = Degrees of freedom; *Vc* = Cramer's V coefficient; RPSG = Repercussion of the Player Substitutions on Goals; FR = Final Result; IRPSG = Impact of the Repercussion of the Player Substitutions on Goals. * $p < 0.05$.

The *ASRs* of the crossover of the RPSG, FR and IRPSG variables are shown in Table 3. The positive impact, regardless of the type of RPSG, implies more cases than expected in matches that end in a win, while the negative impact implies more cases than expected in matches that end in a loss. Likewise, the positive short-term impact is 90.40% win-related, while the negative short-term impact is 90.30% loss-related.

*4.2. MS, FR and IRPSG Variables*

Table 4 shows the degree of relationship between the MS, FR and IRPSG variables. There are significant associations, regardless of MS, with a high strength association.

**Table 3.** *ASRs* of the crossover of the RPSG, FR and IRPSG variables.

| RPSG | IRPSG | Tie | | | Win | | | Loss | | |
|---|---|---|---|---|---|---|---|---|---|---|
| | | *n* | % | *ASRs* | *n* | % | *ASRs* | *n* | % | *ASRs* |
| Immediate | Positive impact | 12 | 50.00 | 0.00 | 25 | 75.80 | 3.70 * | 8 | 24.20 | −3.70 * |
| | Negative impact | 12 | 50.00 | 0.00 | 8 | 24.20 | −3.70 * | 25 | 75.80 | 3.70 * |
| Short-term | Positive impact | 15 | 50.00 | 0.00 | 85 | 90.40 | 10.40 * | 9 | 9.70 | −10.30 * |
| | Negative impact | 15 | 50.00 | 0.00 | 9 | 9.60 | −10.40 * | 84 | 90.30 | 10.30 * |
| Medium-term | Positive impact | 17 | 50.00 | 0.00 | 58 | 89.20 | 8.20 * | 7 | 10.60 | −8.20 * |
| | Negative impact | 17 | 50.00 | 0.00 | 7 | 10.80 | −8.20 * | 59 | 89.40 | 8.20 * |
| Medium–long-term | Positive impact | 8 | 50.00 | 0.00 | 31 | 88.60 | 5.90 * | 4 | 11.40 | −5.90 * |
| | Negative impact | 8 | 50.00 | 0.00 | 4 | 11.40 | −5.90 * | 31 | 88.60 | 5.90 * |
| Long-term | Positive impact | 17 | 50.00 | 0.00 | 82 | 91.10 | 10.20 * | 8 | 8.90 | −10.20 * |
| | Negative impact | 17 | 50.00 | 0.00 | 8 | 8.90 | −10.20 * | 82 | 91.10 | 10.20 * |

Note: *n* = Frequency; RPSG = Repercussion of the Player Substitutions on Goals; FR = Final Result; IRPSG = Impact of the Repercussion of the Player Substitutions on Goals. * *ASRs* > |1.96|.

**Table 4.** Degree of relationship between the MS, FR and IRPSG variables.

| Relationship | $X^2$ | *df* | *p*-Value | *Vc* | Strength |
|---|---|---|---|---|---|
| Tying * FR * IRPSG | 636.875 | 4 | 0.00 * | 0.88 | High |
| Winning * FR * IRPSG | 262.008 | 4 | 0.00 * | 0.51 | High |
| Losing * FR * IRPSG | 262.008 | 4 | 0.00 * | 0.51 | High |

Note: $X^2$ = Chi-square test; *df* = Degrees of freedom; *Vc* = Cramer's V coefficient; MS = Match Status; FR = Final Result; IRPSG = Impact of the Repercussion of the Player Substitutions on Goals. * $p < 0.05$.

The interaction between the MS, FR and IRPSG variables is shown in Table 5. Regardless of MS, the positive impact implies more cases than expected in matches that end in a win. Similarly, the negative impact implies more cases than expected in matches that end in a loss.

**Table 5.** *ASRs* of the crossover of the MS, FR and IRPSG variables.

| MS | IRPSG | Tie | | | Win | | | Loss | | |
|---|---|---|---|---|---|---|---|---|---|---|
| | | *n* | % | *ASRs* | *n* | % | *ASRs* | *n* | % | *ASRs* |
| Tying | Positive impact | 15 | 12.50 | −9.50 * | 103 | 85.80 | 18.00 * | 2 | 1.70 | −7.10 * |
| | Negative impact | 15 | 12.50 | −9.50 * | 2 | 1.70 | −7.10 * | 103 | 85.80 | 18.00 * |
| | No impact | 170 | 100 | 17.50 * | 0 | 0.00 | −10.00 * | 0 | 0.00 | −10.00 * |
| Winning | Positive impact | 5 | 3.00 | −4.20 * | 159 | 97.00 | 5.30 * | 0 | 0.00 | −3.10 * |
| | Negative impact | 49 | 48.00 | 12.90 * | 34 | 33.30 | −16.10 * | 19 | 18.60 | 8.80 * |
| | No impact | 4 | 1.70 | −6.50 * | 232 | 98.30 | 8.00 * | 0 | 0.00 | −4.20 * |
| Losing | Positive impact | 49 | 48.00 | 12.90 * | 19 | 18.60 | 8.80 * | 34 | 33.30 | −16.10 * |
| | Negative impact | 5 | 3.00 | −4.20 * | 0 | 0.00 | −3.10 * | 159 | 97.00 | 5.30 * |
| | No impact | 4 | 1.70 | −6.50 * | 0 | 0.00 | −4.20 * | 232 | 98.30 | 8.00 * |

Note: *n* = Frequency; MS = Match Status; FR = Final Result; IRPSG = Impact of the Repercussion of the Player Substitutions on Goals. * *ASRs* > |1.96|.

### 4.3. ML, FR and MS Variables

Table 6 shows the degree of relationship between the ML, FR and MS variables. There are significant associations between these variables, regardless of ML, with a high strength association.

**Table 6.** Degree of relationship between the ML, FR and MS variables.

| Relationship | $X^2$ | df | p-Value | Vc | Strength |
|---|---|---|---|---|---|
| Home team * FR * MS | 520.144 | 4 | 0.00 * | 0.61 | High |
| Visiting team * FR * MS | 520.144 | 4 | 0.00 * | 0.61 | High |
| Neutral team * FR * MS | 12.000 | 1 | 0.01* | 1.00 | High |

Note: $X^2$ = Chi-square test; *df* = Degrees of freedom; *Vc* = Cramer's V coefficient; ML = Match Location; FR = Final Result; MS = Match Status. * $p < 0.05$.

The *ASRs* of the crossover of the ML, FR and MS variables are shown in Table 7. In terms of the relationship between MS and FR, there are no major differences between playing as home teams or visiting teams when looking at *ASRs*, although the percentage of the home teams winning the matches is somewhat higher.

**Table 7.** *ASRs* of the crossover of the ML, FR and MS variables.

| ML | MS | Tie | | | Win | | | Loss | | |
|---|---|---|---|---|---|---|---|---|---|---|
| | | *n* | % | *ASRs* | *n* | % | *ASRs* | *n* | % | *ASRs* |
| Home team | Tying | 100 | 48.80 | 10.70 * | 58 | 28.30 | −6.20 * | 47 | 22.90 | −3.00 * |
| | Winning | 28 | 9.40 | −7.20 * | 260 | 87.00 | 18.50 * | 11 | 3.70 | −13.50 * |
| | Losing | 30 | 15.20 | −2.90 * | 8 | 4.10 | −14.10 * | 159 | 80.70 | 17.80 * |
| Visiting team | Tying | 100 | 48.80 | 10.70 * | 47 | 22.90 | −3.00 * | 58 | 28.30 | −6.20 * |
| | Winning | 30 | 15.20 | −2.90 * | 159 | 80.70 | 17.80 * | 8 | 4.10 | −14.10 * |
| | Losing | 28 | 9.40 | −7.20 * | 11 | 3.70 | −13.50 * | 260 | 87.00 | 18.50 * |

Note: *n* = Frequency; ML = Match Location; FR = Final Result; MS = Match Status. * *ASRs* > |1.96|.

### 4.4. MS, FR and PST Variables

Table 8 shows the degree of relationship between the MS, FR and PST variables. There is only a significant association between the three variables when the teams are tying, with a low association.

**Table 8.** Degree of relationship between the MS, FR and PST variables.

| Relationship | $X^2$ | df | p-Value | Vc | Strength |
|---|---|---|---|---|---|
| Tying * FR * PST | 54.596 | 22 | 0.00 * | 0.26 | Low |
| Winning * FR * PST | 21.992 | 24 | 0.58 | 0.15 | Low |
| Losing * FR * PST | 20.249 | 24 | 0.68 | 0.14 | Low |

Note: $X^2$ = Chi-square test; *df* = Degrees of freedom; *Vc* = Cramer's V coefficient; MS = Match Status; FR = Final Result; PST = Player Substitutions Time. * $p < 0.05$.

Finally, the interaction between the MS (tying), FR and PST variables is shown in Table 9. When a team is tying, if a player substitution is made in the 60:01 to 65:00 minute, 44.40% of the matches end in a win. However, when a player substitution is made in the 70:01 to 75:00 minute, 69.00% of the matches end in a tie. Similarly, when a player substitution occurs in the 90:01 minute to end of 2nd half, the matches also end in a tie.

### 4.5. Decision Tree with the Following Variables: SC, MS, IRPSG and FR

Figure 2 shows the results of the decision tree designed with the six variables SC, MS, PST, RPSG and IRPSG as independent variables and FR as the dependent variable. The PST and RPSG were eliminated because they did not have a significant influence in this model. The decision tree is composed of 14 nodes shown from top to bottom and from left to right. In this regard, node 0 describes the dependent variable in percentages (tie, win and loss).

**Table 9.** *ASRs* of the crossover of the MS (tying), FR and PST variables.

| MS | PST | Tie | | | Win | | | Loss | | |
|---|---|---|---|---|---|---|---|---|---|---|
| | | *n* | % | *ASRs* | *n* | % | *ASRs* | *n* | % | *ASRs* |
| Tying | 0:00 to end 1st half | 1 | 6.70 | −3.30 * | 7 | 46.70 | 1.90 | 7 | 46.70 | 1.90 |
| | Minute 45:00 to 50:00 | 0 | 0.00 | −3.60 * | 5 | 38.50 | 1.10 | 8 | 61.50 | 3.00 * |
| | Minute 50:01 to 55:00 | 0 | 0.00 | −1.00 | 0 | 0.00 | −0.60 | 1 | 100 | 1.70 |
| | Minute 55:01 to 60:00 | 6 | 37.50 | −0.90 | 5 | 31.30 | 0.50 | 5 | 31.30 | 0.50 |
| | Minute 60:01 to 65:00 | 6 | 33.30 | −1.30 | 8 | 44.40 | 1.90 | 4 | 22.20 | −0.30 |
| | Minute 65:01 to 70:00 | 8 | 40.00 | −0.80 | 7 | 35.00 | 1.00 | 5 | 25.00 | −0.10 |
| | Minute 70: 01 to 75: 00 | 20 | 69.00 | 2.30 * | 5 | 17.20 | −1.10 | 4 | 13.80 | −1.50 |
| | Minute 75:01 to 80:00 | 15 | 60.00 | 1.20 | 2 | 8.00 | −2.10 * | 8 | 32.00 | 0.80 |
| | Minute 80:01 to 85:00 | 21 | 58.30 | 1.20 | 8 | 22.20 | −0.50 | 7 | 19.40 | −0.90 |
| | Minute 85:01 to 90:00 | 14 | 60.90 | 1.20 | 3 | 13.00 | −1.40 | 6 | 26.10 | 0.10 |
| | 90:01 to end 2nd half | 9 | 100 | 3.10 * | 0 | 0.00 | −1.80 | 0 | 0.00 | −1.80 |

Note: *n* = Frequency; MS = Match Status; FR = Final Result; PST = Player Substitutions Time. * *ASRs* > |1.96|.

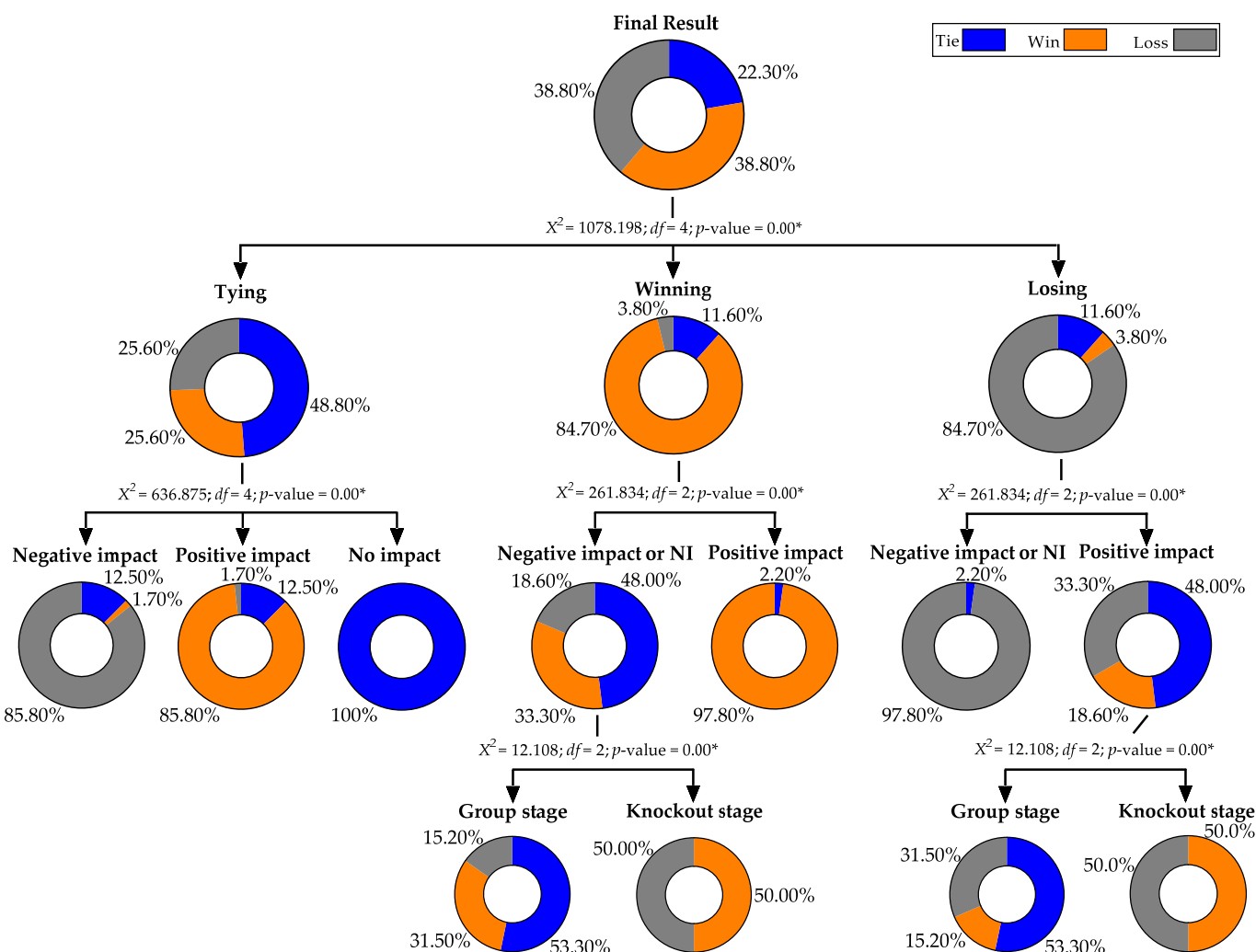

**Figure 2.** Decision tree according to the SC, MS, IRPSG and FR. Note: $X^2$ = Chi-square test; *df* = Degrees of freedom; NI = No Impact. * $p < 0.05$.

Moreover, node 1 indicates that 48.80% of the matches in which a team is tying end in a tie. Node 2 indicates that 84.70% of the matches in which a team is winning end in a win. Node 3 indicates that 84.70% of the matches in which a team is losing end in a loss.

From these nodes are derived other nodes (nodes 4 to 10) that indicate that a win is obtained to a greater extent when a team is tying/winning and also that the impact is positive. On the other hand, when a team is losing and there is a negative impact or no impact, the percentage of loss is very high (97.80%).

Nodes 12 and 14 show that, in the knockout stage, 50% of the matches end in a win and 50% in a loss. When a team is winning and there is a negative impact or no impact, 31.50% of the group stage matches end in a win (node 11). Likewise, when a team is losing and there is a positive impact, only 15.20% of the group stage matches end in a win.

## 5. Discussion

Elite soccer demands positive results [9]. Therefore, the purpose of this study was to describe the relationship/association of player substitution variables with the results of men's UCL matches during the 2018–2019 season. The main results indicated that (1) the positive impact of goal scoring predicted a win when teams were tying or winning; (2) player substitutions were normally made from the 60th minute onwards; (3) the ML did not influence the FR; and (4) a win was obtained to a greater extent when teams that were tying made player substitutions before the 60th minute.

In our study, the short-term positive impact was directly and most strongly related to winning. Amez et al. [27] reported similar results in their study on the UCL and UEFA Europa League. They observed that teams increased the probability of scoring after their 1st and 2nd player substitutions, with this association being less clear during the first three minutes after player substitution. The substituted player, during those first three minutes, is in a process of adaptation to the intensity of the match [27], and the team is in a process of adapting to the tactical changes caused by the player substitution [4,15]. In addition, special care should be taken with player substitutions made in the last few minutes, as a substituted player and team may not have enough time to adapt. Thus, based on the findings obtained, coaches should establish strategies keeping in mind that the impact of the player substitutions on the FR will be evident 5:00 to 10:00 minutes after the player substitution occurs.

In this study, negative impact led to a loss, while positive impact led to a win. Along this line of thought, Rey et al. [5] found that, when teams were tying, coaches made player substitutions earlier than when teams were winning. Teams that scored first had the highest number of wins in Spanish League [26]. Our results point to the importance of the IRPSG when teams were tying during the matches, so coaches should make player substitutions as soon as possible if the team is tying.

Likewise, there was no difference between home teams and visiting teams in the MS and FR relationship. Previous studies on women's elite soccer [34,35] reported that ML in the impact of scoring first did not influence the FR. In the UCL, the ML of the knockout stage did not predict the FR [23]. On the other hand, scoring a goal first, whether it is the home team or visiting team, is important because it increases the probability of winning [26,36]. In contrast, Caballero et al. [37] found that ML had an influence on the FR in the junior men's category. In the Spanish League, home teams had advantages in terms of performance indicators, e.g., shooting [24]. Similarly, Van Dame and Baert [38] indicated that home advantage in the men's UCL is evident when the number of spectators is high and when the home team plays at a higher altitude. In the German League, distance traveled had an effect on home advantage, but this factor has been reduced over time [39]. In term of substitutions, in the national Spanish League, home teams made player substitutions before visiting teams [15]. These findings may be useful for coaches who have a different player substitution strategy depending on the ML.

This study shows that, when teams were winning or losing, the PST did not affect the FR. On the other hand, when teams were tying, the PST did affect the FR. Player substitutions made before the 60th minute were related to a win, while those made after the 70th minute were related to a FR of a tie. These results suggest that, to alter the tie, player substitution must occur before the 70th minute, and that the period from 60:01 to

65:00 minutes showed evidence of more significant results. In this regard, Myers [25] downplays the importance of the minute in which players substitutions are made when teams are either tying or winning. Based on the reported results, coaches can know the PST in which their player substitutions can alter the FR of the match when their team is tying.

Coaches usually made the first player substitution in the period from the 60:01 to the 70:00 minute, the second player substitution from minute 70:01 to 80:01 and the third from minute 80:01 to 90:00. When teams are losing, as the match progresses, the probability of losing becomes greater; therefore, the PST is important for coaches. Bradley [40] also reported that most of the player substitutions occurred between 60 and 85 minute. Gomez et al. [15] indicated that most first and second player substitutions occurred in the period from the 61st to the 90th minute, while third player substitution predominantly occurred in the period from the 76th to the 90th minute. These findings provide coaches with information on substitution patterns that they can integrate into their management approach.

The decision tree served as a confirmatory element of the results presented above, and affirmed that the MS is an indicator to take into account when making player substitutions and that it acts as a predictor variable of the FR. In the UCL, Rey et al. [5] reported that MS was one of the key factors to take into account when studying player substitutions and performance in soccer. On the other hand, positive impact predicted a win when teams were tying and winning, and to a lesser extent when they were losing. Along this line, it has been shown that player substitution creates a new factor in the match, in which the substituted player increases the probability of scoring more goals [41], regardless of the MS. A period of time is required after the player substitutions are made in which the substituted players are in a process of adaptation to the intensity of the game, and the team in a process of adaptation to the tactical changes produced by the player substitutions [27]. It was also observed that winning and facing a negative impact from a player substitution predicted a win to a greater extent than when teams were losing and got a positive impact. Amez et al. [27] pointed out that the probability of scoring was substantially higher if the teams were losing at the time of player substitution. Finally, it should be noted that the differences shown according to the SC could be induced by the rules, i.e., in the knockout stage there must be a winning team that advances to the next round.

## 6. Strengths, Limitations and Proposals for Improvement

This study proposes different contributions for soccer coaches to develop effective player substitution strategies, in order to achieve an improvement in performance and a positive result in matches. The reported results refer to the 2018–2019 UCL, so they cannot be generated for national leagues or for soccer on different continents. Some study limitations are presented, highlighting possible errors derived from the collection of data in the observation sheet, the existence of unobserved factors that mediate the relationship between player substitutions and goal probability, and the fact that the reason for player substitutions was not taken into account. The ability of the coaches and staff to read the flow of the game could have an impact on the results, e.g., a coach with a low level that is incapable of replacing the right players at the right time. A possible improvement would be to use a larger sample, taking into account the different UCL seasons and allowing a more consistent analysis of data. Likewise, other factors not observed in this study could be analyzed, such as the difference in the number of days between matches for each team, taking into account local competitions, as well as the participation of players with their national teams in international competitions. In addition, the specific position of substituted players could be taken into account. It is also proposed to conduct this same study in a season that allows coaches to make up to five player substitutions, with the 2020–2021 season being the first in which coaches had this possibility from the start of the competition.

## 7. Conclusions

The main results indicate that, from the moment a player substitution is made, it takes at least five to ten minutes for there to be an impact on the possibility of scoring a goal and, as a consequence, on the final result. The positive impact of the player substitution predicts a win, regardless of the match status.

In addition, the match status should be taken into account when making player substitutions. In this regard, when the teams are tying, the period from 60:01 to 65:00 is the most appropriate for making substitutions.

There are no differences between home and visiting teams, but there are differences in substitution patterns depending on the competition stage because there must always be a winning team. The first and third player substitutions are made earlier in the group stage, while the second player substitution is made earlier in the knockout stage.

**Author Contributions:** Conceptualization, B.I. and S.J.I.; methodology, J.G.-R. and S.J.I.; formal analysis, B.I. and S.J.I.; investigation, B.I. and J.M.G.-C., resources, S.J.I.; data curation, B.I., J.G.-R. and S.J.I.; writing—original draft preparation, J.M.G.-C.; writing—review and editing, J.M.G.-C., J.G.-R. and S.J.I.; visualization, J.M.G.-C.; supervision, J.G.-R. and S.J.I.; funding acquisition, S.J.I. All authors have read and agreed to the published version of the manuscript.

**Funding:** This study was partially subsidized by the Aid for Research Groups (GR21149) from the Regional Government of Extremadura (Department of Economy, Science and Digital Agenda), with a contribution from the European Union from the European Funds for Regional Development. The author J.M.G.-C. was supported by a grant from the Universities Ministry of Spain and the European Union (NextGenerationUE) "Ayuda del Programa de Recualificación del Sistema Universitario Español, Modalidad de ayudas Margarita Salas para la formación de jóvenes doctores" (MS-01).

**Institutional Review Board Statement:** Not applicable.

**Informed Consent Statement:** Not applicable.

**Data Availability Statement:** Data will be available upon reasonable request to the corresponding author.

**Conflicts of Interest:** The authors declare no conflict of interest.

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
