# Peer review of "How Do Player Substitutions Influence Men’s UEFA Champions League Soccer Matches?"

_applsci, doi:10.3390/app122211371_

Round 1
Reviewer 1 Report
Dear authors,
I would like to express my gratitude regarding the opportunity to review this manuscript.
The study topic is interesting and relevant from the practical application perspective. At this stage the manuscript requires improvements. Below comments and suggestions with line indication:
6 – Please align the text.
17 – Please review the upper and lowercase.
27-28 – Please review the English here and throughout the manuscript.
37 – “play. [5].” – Please correct.
104-106 – Reference to “soccer” is suggested.
122 – Please standardize number´s format (in full or with number – “round of 16” is different compared to “quarterfinals”).
133 – “70: 01” – Please correct.
135 - Overtime – No reference in results (only in table 9) and discussion. Please consider discussing this specific period, it has a duration of thirty minutes and substitutions can have an impact on the soccer game.
142 – “RPSG= GT - PST. “– Please correct.
157 – Please review the hyperlink considering journal template and instructions for authors.
187 – Please insert space.
193 – Please format the “RPSG” column.
202 – Please insert space.
206 – Please format the “MS “column.
216 – Please insert space.
220 – Please format the “ML” column.
229 - Please insert space.
234 – Please format the “MS” and “PST” column.
234 – Please correct “5:00;”.
234 – No reference is observed to “FR”, mentioned in table´s title and note. Please review.
255 – “Neg. impact/NI” and “Negative impact” in the same figure. Please standardize.
262 – “the ML was not influence the FR”. Please review the English here and throughout the manuscript.
295 – “70th” – Please standardize.
308 – “occurred predominantly” or “predominantly occurred”. Please review the English here and throughout the manuscript.
318-319 – A paragraph is suggested.
369 – Please carefully review the references format considering the journal template and instructions for authors. Some examples (please carefully review all references): ref 1 title in uppercase and ref 3 in lowercase (please standardize all references). Please insert journal´s abbreviations. Ref 5 does not have doi (please review all others). 382 “Availabe” – correct. Please review the hiperlinks. 390 “al., e.” Please review and correct.
Author Response
We would like to express our gratitude again to reviewer 1 for the time in reviewing our manuscript and for providing us comments helpful to improve this manuscript quality. We have answered their concerns (all corrections were marked in red and/or change control).
--------------------
In general: In addition to the reviewers' comments, information has been incorporated into the introduction and discussion sections for improvement.
--------------------
Reviewer’ note: 6 – Please align the text.
Authors’ response: Thank you for your suggestion. The text has been aligned (Line 6).
--------------------
Reviewer’ note: 17 – Please review the upper and lowercase.
Authors’ response: Thank you for your suggestion. The term "Contingency Tables" has been written in lower case (Line 17).
--------------------
Reviewer’ note: 27-28 – Please review the English here and throughout the manuscript
Authors’ response: Based on your comments, the English has been revised (Line 27 to 28).
--------------------
Reviewer’ note: 37 – “play. [5].” – Please correct.
Authors’ response: Thank you for pointing out the mistake. It has been corrected (Line 37).
--------------------
Reviewer’ note: Reference to “soccer” is suggested.
Authors’ response: We agree with you. Reference has been made to soccer (Line 120).
--------------------
Reviewer’ note: 122 – Please standardize number´s format (in full or with number – “round of 16” is different compared to “quarterfinals”).
Authors’ response: Thank you for your comment. The format of the numbers has been standardized (Figure 1).
--------------------
Reviewer’ note: 133 – “70: 01” – Please correct.
Authors’ response: Thank you for pointing out the mistake. It has been corrected (Line 148).
--------------------
Reviewer’ note: 135 - Overtime – No reference in results (only in table 9) and discussion. Please consider discussing this specific period, it has a duration of thirty minutes and substitutions can have an impact on the soccer game.
Authors’ response: Thank you for your suggestion. The overtime category has not been included in the inferential analysis of the study because it did not statistically significant (Line 150 to 151).
--------------------
Reviewer’ note: “RPSG= GT - PST. “– Please correct.
Authors’ response: Thank you for your comment. This is a formula (Line 158).
--------------------
Reviewer’ note: 157 – Please review the hyperlink considering journal template and instructions for authors.
Authors’ response: We agree with you. The application used in the study has been referenced according to the journal's instructions (Line 173).
--------------------
Reviewer’ note: 187 – Please insert space.
Authors’ response: Thank you for pointing out the mistake. It has been corrected (Line 204).
--------------------
Reviewer’ note: 193 – Please format the “RPSG” column.
Authors’ response: Thank you for your comment. The “RPSG” column has been formatted (Table 3).
--------------------
Reviewer’ note: 202 – Please insert space.
Authors’ response: Thank you for pointing out the mistake. It has been corrected (Line 218).
--------------------
Reviewer’ note: 206 – Please format the “MS “column.
Authors’ response: Thank you for your comment. The “MS” column has been formatted (Table 5).
--------------------
Reviewer’ note: 216 – Please insert space.
Authors’ response: Thank you for pointing out the mistake. It has been corrected (Line 232).
--------------------
Reviewer’ note: 220 – Please format the “ML” column.
Authors’ response: Thank you for your comment. The “ML” column has been formatted (Table 7).
--------------------
Reviewer’ note: 229 - Please insert space.
Authors’ response: Thank you for pointing out the mistake. It has been corrected (Line 245).
--------------------
Reviewer’ note: 234 – Please format the “MS” and “PST” column.
Authors’ response: Thank you for your comment. The “MS” and “PST” columns have been formatted (Table 9).
--------------------
Reviewer’ note: 234 – Please correct “65:00;”.
Authors’ response: Thank you for pointing out the mistake. It has been corrected (Table 9).
--------------------
Reviewer’ note: 234 – No reference is observed to “FR”, mentioned in table´s title and note. Please review.
Authors’ response: Thank you for your comment. In this case, "FR" refers to tie, win and loss as in tables 3, 5 and 7.
--------------------
Reviewer’ note: 255 – “Neg. impact/NI” and “Negative impact” in the same figure. Please standardize.
Authors’ response: Thank you for your suggestion. The term “Negative impact” has been used (Figure 2).
--------------------
Reviewer’ note: 262 – “the ML was not influence the FR”. Please review the English here and throughout the manuscript.
Authors’ response: Based on your comment, the English has been revised (Line 279).
--------------------
Reviewer’ note: 295 – “70th” – Please standardize.
Authors’ response: Thank you for pointing out the mistake. It has been corrected (Line 316).
--------------------
Reviewer’ note: 308 – “occurred predominantly” or “predominantly occurred”. Please review the English here and throughout the manuscript.
Authors’ response: Based on your comment, the English has been revised (Line 329).
--------------------
Reviewer’ note: Please carefully review the references format considering the journal template and instructions for authors. Some examples (please carefully review all references): ref 1 title in uppercase and ref 3 in lowercase (please standardize all references). Please insert journal´s abbreviations. Ref 5 does not have doi (please review all others). 382 “Availabe” – correct. Please review the hiperlinks. 390 “al., e.” Please review and correct.
Authors’ response: Thank you for your comments. All references have been reviewed.
Kind regards.
Reviewer 2 Report
Please check the attached documents.

Author Response
We would like to express our gratitude again to reviewer 2 for the time in reviewing our manuscript and for providing us comments helpful to improve this manuscript quality. We have answered their concerns (all corrections were marked in red and/or change control).
Scientific evidence has shown that for the sample of matches analyzed and in the competition analyzed, a minimum time is necessary for substitutions to be effective. Therefore, the authors have highlighted the inclusion of the analyzed competition in the title of the manuscript. We thank them for their comments.
--------------------
In general: In addition to the reviewers' comments, information has been incorporated into the introduction and discussion sections for improvement.
--------------------
Reviewer’ note: Line 54 - I agree the authors opinions, to change the tactical behavior of the teams is the main reason for making player substitutions. However, I believe that the match congestions in modern soccer, including regular domestic league matches and major cup competitions, has an impact on player substitution and turnover.
Authors’ response: Thank you for your suggestion. This information has been incorporated and cited in the manuscript (Line 62 to 69).
Julian, R.; Page, R.M.; Harper, L.D. The Effect of Fixture Congestion on Performance During Professional Male Soccer Match-Play: A Systematic Critical Review with Meta-Analysis. Sports Medicine 2021, 51, 255–273, doi:https://doi.org/10.1007/s40279-020-01359-9.
--------------------
Reviewer’ note: Although the authors mentioned the existence of unobserved factors in limitations, I think that the number of days between matches and difference between them in each team could be observed.
Authors’ response: We agree with you. The difference in the number of days between matches per team has been added as a future study proposal (Line 361 to 365).
--------------------
Reviewer’ note: I suggest that to present what variables cannot be observed and what variables can be easily measured by an external survey. Of course, it will not be easy to measure the reasons for player substitutions. All variables used to this study should have reasons for their selection. Conversely, variables that were not used must have a reason for not being used.
Authors’ response: Thank you for your suggestion. The reasons for the selection of the variables analyzed in this study has been indicated (Line 71 to 74).
--------------------
Reviewer’ note: Line 107 - The purpose of this study was to identify the relationship between player substitutions and match results in the 2018-2019 season. This expression does not distinguish between independent and dependent variables. It would be more desirable to describe the relationship/association of player substitution variables with match results.
Authors’ response: Thank you for your suggestion. The purpose of the study has been rewritten (Line 13 to 14; line 121 to 123; line 276 to 277).
--------------------
Reviewer’ note: Home-and-away results may also depend on the distance traveled.
Authors’ response: Thank you for your comment. This information has been discussed in the manuscript (Line 309-310).
Beckmann, N. Statistical influence of travelling distance on home advantage over 57 years in the men’s German first soccer division. German Journal of Exercise and Sport Research 2021, 10.1007/s12662-021-00787-7, doi:10.1007/s12662-021-00787-7.
--------------------
Reviewer’ note: The results of this study are only the results of this season’s competition and cannot be generalized to domestic league or soccer on different continents. This should be noted in the study limitations. The ability of the coach and staff to read the flow of the game will also have an impact. Therefore, if the level of coach is low and the right players cannot be replaced at the right time, the results may vary.
Authors’ response: Thank you for your suggestions. In limitations section, the following suggestions have been incorporated: the results cannot be generalized to other competitions (Line 352 to 353); the ability of the coach and staff could have an effect on results (Line 357 to 359).
Kind regards.
Round 2
Reviewer 1 Report
Dear authors,
Thank you for considering my suggestions and incorporating them into the manuscript.
Below suggestions related to this last version (v2), with line indication.
L14 – Please describe UCL in full, it is the first appearance in the manuscript.
L150-152 – Please correct and improve the English “The overtime category did not included in the inferential analysis of this study because it did not statistically significant”. Also, this sentence is questionable, the non-inclusion of this category may represent a limitation or suggestion for future research”
201 – Please review and standardize the table lines (line above 164 different from above 201). Same in table 9 (line 250).
236 – Please review the text in bold and normal format in table 7 and standardize.
273 – In the same line the text presented is “Negative impact” and “Negative impact/NI”, which seems to represent the same. Consequently, standardization is suggested.
Please carefully review and correct all the references according to journal instructions for authors (https://www.mdpi.com/journal/applsci/instructions), for example:
· - Abbreviated Journal Name;
· - Title of the article standardized (upper and lowercase – for example refs 1 and 2 different).
And carefully review other details in all the 41 references.
A careful review/reading after considering this round 2 is suggested, with special attention to details and English.
Author Response
We would like to express our gratitude again to reviewer 1 for the time in reviewing our manuscript and for providing us comments helpful to improve this manuscript quality. We have answered their concerns (all corrections were marked in red and/or change control).
--------------------
Manuscript writing: An experienced native translator reviewed the English of the entire manuscript.
--------------------
Reviewer’ note: L14 – Please describe UCL in full, it is the first appearance in the manuscript.
Authors’ response: We agree with you. Therefore, UCL has been described in its entirety (Line 14).
--------------------
Reviewer’ note: L150-152 – Please correct and improve the English “The overtime category did not included in the inferential analysis of this study because it did not statistically significant”.
Authors’ response: Thank you for your suggestion. The English wording has been revised (Line 152 to 153).
--------------------
Reviewer’ note: 201 – Please review and standardize the table lines (line above 164 different from above 201). Same in table 9 (line 250).
Authors’ response: Based on your comment, the lines of all tables in the manuscript have been revised and standardized.
--------------------
Reviewer’ note: 236 – Please review the text in bold and normal format in table 7 and standardize.
Authors’ response: Following your comment, the bold text has been corrected and standardized (Table 7).
--------------------
Reviewer’ note: 273 – In the same line the text presented is “Negative impact” and “Negative impact/NI”, which seems to represent the same. Consequently, standardization is suggested.
Authors’ response: Thank you for your suggestion. The same information has been standardized in the text and in Figure 2: "negative impact or no impact" (Line 273). NI means no impact (as is clarified in the legend below the figure), so this is not the same. It also clarified in the text what does it means (variables section).
--------------------
Reviewer’ note: Please carefully review and correct all the references according to journal instructions for authors.
Authors’ response: Thank you for your comment. The references have been corrected according to the authors' instructions.
Kind regards